# Identification of Runs of Homozygosity Islands and Genomic Estimated Inbreeding Values in Caqueteño Creole Cattle (Colombia)

**DOI:** 10.3390/genes13071232

**Published:** 2022-07-12

**Authors:** Alejandra M. Toro-Ospina, Ana C. Herrera Rios, Gustavo Pimenta Schettini, Viviana H. Vallejo Aristizabal, Wellington Bizarria dos Santos, Cesar A. Zapata, Edna Gicela Ortiz Morea

**Affiliations:** 1Amazonian Research Center CIMAZ-MACAGUAL, Laboratory of Agrobiotechnology, University of the Amazon, Florencia 180002, Colombia; ana.herrera@iudigital.edu.co (A.C.H.R.); vallaristy@gmail.com (V.H.V.A.); cesaruniamazonia@gmail.com (C.A.Z.); omorea1010@gmail.com (E.G.O.M.); 2Science and Humanities Faculty, Digital University Institute of Antioquia, IUDigital, Medellin, Antioquia 50010, Colombia; 3Department of Animal and Poultry Science, Virginia Polytechnic Institute and State University, Blacksburg, VA 24061-0002, USA; gschettini@vt.edu; 4School of Agricultural and Veterinary Sciences (FCAV), São Paulo State University (UNESP), Jaboticabal, Sao Paulo 14884-900, Brazil; santwellingtonb@gmail.com

**Keywords:** cattle native, inbreeding, tropical

## Abstract

The Caqueteño Creole (CAQ) is a native breed of cattle from the Caquetá department (Colombia), adapted to tropical conditions, which is extremely important to production systems in those regions. However, CAQ is poorly studied. In this sense, population structure studies associated with runs of homozygosity (ROH) analysis would allow for a better understanding of CAQ. Through ROH analysis, it is possible to reveal genetic relationships between individuals, measure genome inbreeding levels, and identify regions associated with traits of economic interest. Samples from a CAQ population (*n* = 127) were genotyped with the Bovine HD BeadChip (777,000 SNPs) and analyzed with the PLINK 1.9 program to estimate FROH and ROH islands. We highlighted a decrease in inbreeding frequency for FROH 4–8 Mb, 8–16 Mb, and >16 Mb classes, indicating inbreeding control in recent matings. We also found genomic hotspot regions on chromosomes 3, 5, 6, 8, 16, 20, and 22, where chromosome 20 harbored four hotspots. Genes in those regions were associated with fertility and immunity traits, muscle development, and environmental resistance, which may be present in the CAQ breed due to natural selection. This indicates potential for production systems in tropical regions. However, further studies are necessary to elucidate the CAQ production objective.

## 1. Introduction

Cattle are an important livestock species, raised for several purposes, such as meat, milk, leather, and traction. *Bos taurus taurus* and *Bos taurus indicus* subspecies are present throughout the American continent, characterized by their breed diversity due to selection and introgression effects [1]. Furthermore, the genetic diversity and adaptation of these subspecies promoted genomic changes in their DNA, which led to the formation of new breeds. Colombia is a Latin American country with the highest number of Creole cattle breeds adapted to the humid tropics. Creole breeds display resilience and high reproductive indexes; these breeds also are well-adapted to prolonged periods of drought and can utilize low-quality forage [2]. The Criollo Caqueteño (CAQ) breed is found in the Department of Caquetá (Colombia). It is a medium-height animal (from 125 to 140 cm), with a rectilinear head profile, medium-sized ears, and pink mucous membranes; the color of the coat can be red or bay, presenting short and thin hair [3]. At the adult age, females weigh approximately 402.2 kg, and males reach nearly 612 kg; they are rectangular and cylindrical animals, traits from beef cattle breeds. The CAQ breed is adapted to tropical conditions and used in extensive grazing systems. However, CAQ does not have a defined selection objective due to its low population size and the lack of information on its productive and reproductive records. At present, the CAQ breed comprises only a few individuals, which is a concern for Creole breed associations. In this context, breed conservation projects and restoration have developed in the region.

There is still a lack of information on the genetic diversity and population structure of the CAQ breed and other Colombian Creole breeds. Colombia has eight Creole cattle breeds, of which only a few (Blancorejinegro, Romosinuano, and Harton del Valle) have genealogical and phenotypic information available [4]. Delgado et al. [5] studied the relationship between 26 Creole cattle breeds from 10 Latin American countries using microsatellites and identified that Creole cattle retain high levels of genetic diversity. It was also highlighted that most of the studied breeds present genetic variability, despite the small sizes of their populations. It was possible to assess their population structure and productive potential based on that information. However, studies at the genome level are limited in Creole breeds due to a lack of commercial interest and resources to carry out scientific research. In 2007, the bovine genomics era began, which permitted genomic studies using high-density SNP chips to be conducted to explore the genome and investigate genetic diversity and population structure [6]. Thus, it was possible to better understand the genetic architecture of populations through population structure analysis and access its history through linkage disequilibrium (LD) and runs of homozygosity (ROH).

The ROH is a continuous homozygous segment in the genome that is widely used in livestock to characterize population structure and demographic history [7]. Additionally, ROH analysis can help to reveal the genetic relationships between individuals by measuring genomic inbreeding levels and identifying regions with a high frequency of homozygosity (ROH islands) associated with traits of economic interest [8]. Several studies involving native breeds from different countries have used ROH to find the history of their genome; for example, a native Chinese breed [9], the Blanco Orejinegro breed in Colombia [10], and native Italian dairy breeds [11]. ROH islands could also be applied to identify candidate genes associated with relevant phenotypes [12] since there is an expectation that these homozygous regions are close to or within quantitative trait loci undergoing animal selection [13,14]. Thus, the ROH island mapping of a population under selection allows for the genetic variants associated with economically important traits to be identified [15]. Based on that, these analyses could determine whether the natural selection and the environmental conditions faced by the CAQ breed promoted the establishment of potential regions of homozygosity associated with adaptive traits. This will contribute to the development of livestock in the tropical region. The present study aimed to investigate the ROH genomic inbreeding level and identify ROH islands along the CAQ breed genome.

## 2. Materials and Methods

### 2.1. Animals and Genotypes

The CAQ population evaluated in this study comprised a small number of individuals belonging to the nucleus protected by the University of the Amazon, Macagual campus, in the city of Florencia, Caquetá, Colombia. The population consisted of 127 animals (18 males and 109 females) (Figure 1), with the existing population being of the CAQ Creole breed. There are no specific selection or conservation programs for the CAQ breed.

The database was composed of the genotypes of 127 Creole animals genotyped with the BovineHD BeadChip 777,000 SNPs (Illumina, San Diego, CA, USA). Assuming an MAF < 0.01, the PLINK program was applied for quality control of the genotypes [8,12,16], a call rate for SNPs < 0.90, and a call rate for samples < 0.98.

### 2.2. Genetic Structure of the Population

Genomic analysis of the population structure indicates the proportion of the total genetic variability represented by the associated principal component (PC), obtained by the *plink-pca* command line of the PLINK program [17].

Furthermore, to verify the genetic dissimilarity within the population, genetic statistics measures such as identical by descent (IBD) and Wright’s F were obtained to assess individual inbreeding. The IBD estimates for all pairs of individuals; this measure detects sample contaminations, swaps, duplications, pedigree errors, and unknown familial relationships. The estimates of pairwise IBD find pairs of individuals who look too similar, i.e., more than we would expect by chance in a random sample [17].

For IBD calculation, first, the identity by status (IBS) needs to be determined between individual pairs, which reflects whether individuals share zero, one, or two alleles at each SNP. The IBD estimates the probability that the shared alleles were inherited from a common ancestor. The variable π, where π = P (z = 1) 2+ P (z = 2), represents the proportion of shared IBD alleles, and Z represents the IBD status of 0, 1, and 2 [17].

### 2.3. Runs of Homozygosity Detection

The ROH was calculated with the PLINK 1.9 program [17] using a specific-length sliding window. The following parameters were applied: (a) 30 or more consecutive homozygous SNPs; (b) 1 heterozygous per window; (c) 0 missed calls in homozygous windows; (d) minimum proportion of homozygous windows in which each SNP occurs, 0.05; (e) the minimum ROH length was set at 1000 kb; (f) the minimum required SNP density was 1 SNP per 50 kb, and (g) the maximum gap between two consecutive SNPs was set at 500 kb. In addition, the ROH were analyzed in different size classes (ROH1–2 Mb, ROH2–4 Mb, ROH4–8 Mb, ROH8–16 Mb, and ROH > 16 Mb).

The genomic inbreeding from ROH (F_ROH_) accurately detects autozygosity and predicts its current percentage since it can assess a common ancestor even 50 generations previous. Additionally, the F_ROH_ has been used when there is a lack of genealogical information on the population [18,19].

The inbreeding coefficient based on the runs of homozygosity (F_ROH_) of each animal was calculated as follows [19,20]:$$\mathrm{FROHi}=\frac{\;\sum \mathrm{tlength}\;\left(\mathrm{ROHt}\right)}{{\mathrm{Length}\;}_{\mathrm{total}}}$$
where t = the number of each individual’s ROH multiplied by the average length of ROHs, with the total bovine genome length corresponding to 2,612,820 kb. [21]. ROH segments were identified using the PLINK 1.07 software [17] for each F_ROH_ (1–2 Mb, 2–4 Mb, 4–8 Mb, 8–16 Mb, and >16 Mb).

### 2.4. ROH Islands

The ROH islands were calculated with the PLINK 1.9 program [17] using the *--homozyg-group--* command to identify ROH hotspots/islands with a higher frequency in shared regions in the population. The shared ROH regions can be defined as genomic regions with reduced genetic diversity, presenting high homozygosity around the specific site. It is possible to identify the aims of the positive and intense selective selection [14]. These regions were similar among individuals in which homozygote regions were identified. The ROH segments shared by more than 20% of the individuals in the population were considered common regions. In the file for genes of the bovine genome UMD3.1, the ENSEMBL gene IDs were used to perform a functional annotation analysis using the Database for Annotation, Visualization, and Integrated Discovery (DAVID) v6.8 tool [22], and Gene Ontology (GO) Terms and KEGG pathways with a *p*-value < 0.05 were selected.

The experiment was carried out by article 2 of agreement No. 027 of 2020 of the Faculty of Agricultural Sciences, in accordance with the guidelines of the Ethics, Bioethics and Animal Welfare Committee of the University of the Amazon, code CEBBA 129, 2021.

## 3. Results

### 3.1. The Genetic Structure of CAQ Exhibits Low Population Stratifications

In the principal component analysis of the CAQ breed (Figure 2), we found that PC1 explained 24% of the variation. Stratification of the CAQ population was low, forming two clusters. For PC2, a larger cluster was observed, explaining 12% of the variance. Both PCs exhibited genetic variability and indicated the low stratification of the CAQ population. Possibly, the studied animals had suffered from a recent introgression of animals crossed with CAQ, which led to higher genetic variability in the current population.

The average inbreeding coefficient F in the CAQ population was 0.054, with a maximum F of 0.261. IBD analysis was also conducted, aiming to confirm the inbreeding coefficient of the CAQ population. However, the individual pairwise IBD calculations did not identify any sample duplication (pairwise IBD > 0.95), although the analyzed markers identified individuals in the population with genetic similarities of higher than 50%, suggesting a first-degree relationship between specific individual pairs. The mean IBD score in the CAQ population was 0.071. In addition, there were 27 animals with a maximum IBD score of 0.50 and an animal pair with max IBD scores of 0.63 and 0.65.

These PC results indicate that the CAQ population genetically exhibits low population stratification due to the different lineages and possible blood mixing between animals crossed with CAQ, allowing for the population to diversify. The obtained inbreeding coefficients indicate control in current matings, allowing for the population to open up despite its size, which would enable the implementation of a targeted mating strategy and increase the number of individuals in the population.

### 3.2. Genomic Distribution of Runs of Homozygosity

The distribution of ROH was identified in the 127 analyzed individuals, with 6474 ROH segments (Table 1). The average size of each segment was 2.20 MB, ranging from 1 to 17.6. Based on the identified sizes, we can conclude that 98.97% of the ROH segments are short and possibly underwent fragmentation by recombination across generations, indicating that the highest proportion of the genome covered by ROH in the CAQ breed is less than 8 Mb.

The ROH per chromosome was higher for BTA1 (402 segments) and BTA3 (350 segments) and decreased with the increase in chromosome size. Regions were identified on 26 autosomes, indicating the presence of autozygosity in the studied genome. The highest value, considered in autozygosity, was observed for chromosome 3 (0.075%), followed by chromosomes 15 (0.0708%) and 20 (0.067%).

### 3.3. Genomic Inbreeding Coefficient (F_ROH_)

One hundred and twenty-seven animals of the CAQ population were analyzed; however, higher average inbreeding was observed for the F_ROH_ < 2 Mb (16%) and F_ROH_2–4 (12%) classes (Table 2) in 15 animals, indicating ancient inbreeding. This percentage was probably influenced by the founders, in which some animals in the recent population still share ROH segments with their ancestors. A decrease in the inbreeding percentage was observed for the F_ROH_4–8 Mb, F_ROH_8–16, and F_ROH_ > 16 Mb classes, indicating control of inbreeding in recent matings. Only 15 animals of the Creole population were highly homozygous at the ROH1–2 Mb distance compared to the total analyzed population. This finding may be because the CAQ population was not improved without selection pressure, as a strong ancient relationship was only observed among 15 animals, which could be attributed to the founder animal lineage.

### 3.4. Gene Characterization in ROH Islands

Genomic regions with the highest frequency of ROH occurrence are called ROH hotspots/islands. Regions present in more than 20% of the individual population are considered common regions (red dots in Figure 3) and also can harbor genes associated with important production traits in cattle. The genomic regions were located on chromosomes BTA3 (position: 29615176-98903642), BTA5 (62276336-87155753), BTA6 (38632996-116998306), BTA8 (65897661-105645859), BTA16 (43074568-454224), and BTA22 (1174419- 61379134). BTA20 harbored four ROH hotspots (20114020-63708551, 38347984-40037991, 12658291-51430813, and 10769304-63708551) (Figure 3).

A total of 22 genes enriched in 14 biological processes (BP) were found. Ten of them were significant (*p* < 0.05) (*SPEF2, TUT7, MAGI3, PDE4D, PRLR, RSBN1*, *PEX14, SEC61G, PHTF1,* and NUP54). The genes were related to the cellular inflammatory response (GO:0006954) and immune response (GO:0006955), and genes were associated with the positive regulation of biological processes (GO:0032729) and fertility (GO:0007288; GO:0007171). A single significant (*p* ≤ 0.05) enriched KEGG pathway was identified: the pathway for basic and acidic residues (bta04146), related to cytoskeletal actin filament and muscle functions.

## 4. Discussion

### 4.1. Population Structure of the CAQ Population Identified Genetic Variability

The first two principal components explained more than 36% of the total genetic variation, suggesting genetic variability in the CAQ population. Population structure analysis is an essential tool for understanding a population’s history and how the animals contribute to the genetic variability in a population, promoting the emergence of genetic subgroups when new genetic material is input. According to Price et al. [23], population stratification studies help to identify lineages (genetic substructures) and have higher accuracy in the analyses of genomic data. A molecular characterization study using microsatellites in the CAQ population identified genetic variability despite the small effective population size [4]. However, the existing CAQ population consists of approximately 130 individuals, with ten sires being used for matings. The PC suggests that different strains possibly have relevant differences between sires.

The population structure studies by molecular techniques (microsatellites) were carried out in some Colombian Creole breeds. Barrera et al. [4] reported an average inbreeding coefficient of 0.14 for a population of 80 Caqueteño Colombian Creole animals. Possibly, using previous information, the breeders controlled mating in the breed. After 15 years, the present study demonstrated that the inbreeding coefficient in the CAQ population had considerably decreased. Studies involving other Creole breeds reported similar inbreeding coefficients, such as the Romosinuano breed with F = 0.051 [2] and the Casanareño Creole breed with F = 0.08 [24]. These PC results indicate that the CAQ population genetically exhibits low population stratification due to different lineages and possible blood mixing between animals crossed with CAQ, allowing for the population to diversify. The obtained inbreeding coefficients indicate control in current matings, allowing for the population to open up despite its size, which would enable the implementation of a targeted mating strategy and increase the number of individuals in the population.

### 4.2. Runs of Homozygosity (ROH) Characteristics

We identified a higher proportion of short ROH segments in the CAQ genome. Similar results have been reported by Zambrano et al. [25] for a population of Holstein cattle from the Nariño Department, in which short segments accounted for approximately 96% of all identified ROH, indicating the likely low recent selection in the studied population. In this way, short ROH have been associated with ancient inbreeding [25].

Recent genomic studies on the Blanco Orejinegro Creole breed in Colombia showed that the total number of small segments identified in the population accounted for approximately 20.54% of all ROH that were detected in the analysis, indicating a significant ancient inbreeding [10]. In other studies carried out in South America on breeds such as Nelore and Gyr, the genome of these breeds was mainly composed of a large number of short segments that accounted for 78% of all ROH [26]. For the Gyr population, a proportion of its genome (183.6 Mb), covered by an ROH of a shorter length (ROH1–2 Mb), was also evaluated [8]. The identification of ROH in the bovine genome provides information on the specific regions that could be related to selection events. Furthermore, this approach improves the accuracy of estimated inbreeding coefficients and evaluates the genes related to traits of economic interest that are fixed in the population across generations [15] and encouragement in the creole breeds [10]. The distribution of ROH obtained in the present study suggests the higher contribution of remote inbreeding to autozygosity in this CAQ population, since there was a high prevalence of short ROH in the genome. These considerations agree with the results of Sölkner et al. [27] and Howrigan et al. [18].

### 4.3. Identification of Candidate Genes within Runs of Homozygosity

The overlapping of ROH regions of high frequency that were shared among animals suggests a signal from the ROH islands, and these regions were subjected to a gene enrichment analysis (Figure 3). Three candidate genes were identified on BTA3, where the *PHTF1* gene was associated with the fertility index in Holstein cattle [28]. On the other hand, the identified *RSBN1* and MAGI3 genes have been related to immunological processes and disease resistance [29]. On chromosome BTA6, the *CXCL11* gene was identified and has been associated with immune response during pathological processes, including inflammation and autoimmune and infectious diseases in different cattle populations [30]. Moreover, the *NUP54* gene was related to body fat deposition and muscle development [31]. *SCARB2* gene was associated with several functions in energy metabolism, including the synthesis and transport of energy, lipids, and activity in large organs [32]. On BTA8, the *TUT7* gene was linked to immune response cells [33], and the *GAS1* gene was associated with morphological traits such as capacity and body depth, chest width and angularity, and teat length in Chinese Holstein cattle [34]. 

On BTA22, the *SEC61G* gene was associated with musculoskeletal development and growth [35]. According to Hardie et al. [36], the *SEC61G* gene in Holstein cows is related to body size and feed efficiency during the lactation period. The identified genes are possibly associated with traits of economic interest related to fertility, immunity, and body size. These traits are consistent with the high rates of fertility, precocity, and hardiness observed in the CAQ breed [37].

In the CAQ population, four ROH hotspots located on BTA20 also were identified, with two related to productive traits. The *PIK3R1* gene has been associated with an increased milk protein fraction at peak lactation [38]. The *PDE4D* gene has been reported to increase meat marbling in studies on Nellore and Hanwoo cattle [39]. Genes related to immunity and environmental resistance have been identified on BTA20; for example, the candidate gene *CD180* is associated with the animal’s ability to deal with pathogens in Brahman and Angus cattle [40]. Furthermore, the candidate genes *SPEF2* and *PRLR* have been linked to the SLICK phenotype, which has an essential effect on coat length in cattle and can affect cattle thermoregulation in tropical climates [41]. A study in the Senepol breed identified the SLICK phenotype, which was associated with a mutation in the *PRLR* gene that resulted in a short coat. Senepol cattle are one of the few *Bos taurus* breeds that can tolerate heat, possibly due to this phenotype [42,43]. The *PEX14* and *CORT* genes located on BTA16 have been linked to environmental adaptation and the response to heat stress [44,45]. The CAQ breed is well adapted to harsh environments and displays high resistance to infections and diseases, besides its reproductive and productive potential, which are crucial traits for a targeted selection establishment.

The CAQ breed faces adverse environmental conditions, and natural selection has allowed for it to fix the crucial traits for tropical regions in Colombia. Furthermore, Colombian Creole breeds have become relevant due to their genetic potential and significant traits for the Colombian tropics, such as their high fertility and parasite resistance [46]. In addition, these breeds can be kept on poor-quality pastures compared to other commercial breeds. Some of the genes identified in the ROH islands have been used for selection in different commercial cattle breeds to rear animals with shorter coats, improving thermotolerance in humid tropical climates [47]. Currently, studies should be conducted with native breeds, since cattle species face challenges every day, such as those associated with global climate changes that affect animal production. The CAQ breed shows a low tendency to express genes associated with essential traits within this context. However, the CAQ population could shortly be an alternative for production systems in the Colombian humid tropics.

## 5. Conclusions

FROH is an adequate method to analyze inbreeding in the nonexistent pedigree population. The results pointed to recent autozygosity but low inbreeding based on FROH. It is possible that some candidate genes previously related to environmental resistance, immunity, and fertility are present in ROH hotspots at a low frequency in the CAQ population, especially on chromosomes BTA3, BTA4, BTA6, BTA8, and BTA16. In addition, our results suggest that natural selection has at least partially played a role, among other factors, in shaping the genomic patterns of ROH in the CAQ population and provides valuable information for understanding the genetic basis of relevant traits. The findings will serve as a basis for more population studies that can promote the conservation of the CAQ and improve its productive traits in the future. 

## Figures and Tables

**Figure 1 genes-13-01232-f001:**
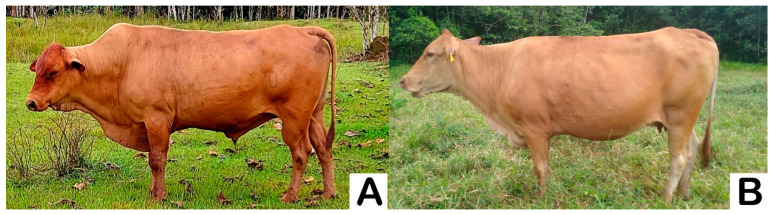
Specimens of Caqueteño Creole breed: male (**A**) and female (**B**).

**Figure 2 genes-13-01232-f002:**
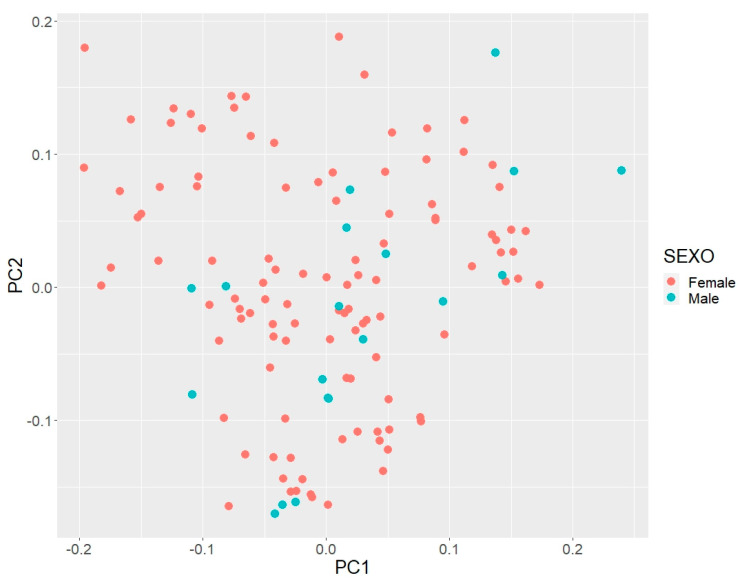
PCA scatter-plot of the first two principal components (PC) of Caqueteño Creole population, where red dots represent females and blue dots represent males.

**Figure 3 genes-13-01232-f003:**
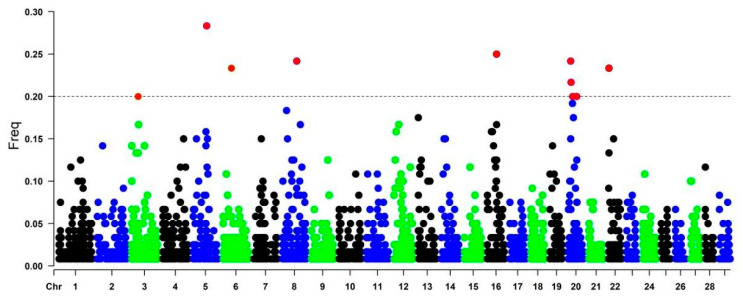
Manhattan plot of the distribution of ROH islands in the CAQ population. The X-axis represents the distribution of ROH along the genome, and the Y-axis shows the frequency (%) of overlapping ROH shared among samples. Red dots above the cutoff value are possible ROH islands.

**Table 1 genes-13-01232-t001:** Distribution of runs of homozygosity (ROH) of different lengths (ROH < 2 Mb, ROH2–4 Mb, ROH4–8 Mb, ROH8–16 Mb and ROH > 16 Mb).

Class	ROH (N)	ROH (%)	Mean	Min	Max	Standard Deviation
(ROH < 2 Mb)	4067	62.82	1.386	1	1.99	0.28
(ROH2–4 Mb)	1752	27.06	2.750	2	3.99	0.55
(ROH4–8 Mb)	589	9.09	5.306	4	7.89	1.00
(ROH8–16 Mb)	65	1.01	9.848	8	14.23	1.80
(ROH > 16 Mb)	1	0.015	17.26	17.26	17.26	17.26
Total ROH	6474	100	2.20	1	17.26	1.50

**Table 2 genes-13-01232-t002:** Genomic inbreeding coefficients based on runs of homozygosity (F_ROH_) for different lengths (1–2 Mb, 2–4 Mb, 4–8 Mb, 8–16 Mb and > 16 Mb).

Variable	N	Mean	Standard Deviation	Sum	Minimum	Maximum
F_ROH_ < 2	15	0.16348	0.04702	2.45226	0.10206	0.24563
F_ROH_ 2–4	15	0.12847	0.04268	1.92711	0.06728	0.21083
F_ROH_ 4–8	15	0.09731	0.05168	1.45964	0.04909	0.24458
F_ROH_ 8–16	14	0.02650	0.02937	0.37098	0.00328	0.09405
F_ROH_ > 16	2	0.00683	0.0000974	0.01367	0.00676	0.00690

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
