# Peer review of "Identification of Runs of Homozygosity Islands and Genomic Estimated Inbreeding Values in Caqueteño Creole Cattle (Colombia)"

_genes, 2022, doi:10.3390/genes13071232_

Round 1
Reviewer 1 Report
In the manuscript, genes-1790508 by Toro-Ospina et al., authors have studied the Caqueteno Creole (CAQ), native cattle from the Caqueta department, Columbia for the identification of runs of homozygosity (ROH) and genomic estimated breeding values. They have used a population sample of 120 animals and genotyped with Bovine HD Bead Chips followed by estimation of FROH and ROH island using PLINK 1.9 program. Overall, the manuscript data is inline with the conclusions/interpretation. There are some issues with the structure/organization of manuscript that would help for the better readability.
1. Abstract- The gap/lacunae and the rationale are not clear in the abstract.
2. Flow of information in the abstract in not well organized. It requires revision.
3. Line 14. “ROH analyzes can help reveal genetic relationships……”. ROH analyzes???
4. Line 21-22. “We found genomic regions on chromosomes BTA3, BTA5, BTA6, BTA8 and BTA16”. What kind of genomic regions?? This sentence needs to be revised.
5. Line 147. Please revise the title of this subheading of the result section. In current form it is not informative. For instance, "Genetic population structure of CAQ exhibits low population stratifications.
6. Figure 2 ligand is not explanatory. Please consider explaining the graph components.
7. Line 170. Please consider revising this title heading. "Runs of homozygosity" is not informative.
8. Line 185. Similar to the above points, please consider revising the title of the heading.
9. Both Results and Discussion sections have used the same heading titles. For the sale of readability and understanding, I would recommend using some informative message containing titles for these heading.
Author Response
Thank for the suggestions on the manuscript. The paragraphs that were revised are highlighted in yellow.
The version revised manuscript has undergone English language editing by MDPI.
Comments and Suggestions for Authors
In the manuscript, genes-1790508 by Toro-Ospina et al., authors have studied the Caqueteno Creole (CAQ), native cattle from the Caqueta department, Columbia for the identification of runs of homozygosity (ROH) and genomic estimated breeding values. They have used a population sample of 120 animals and genotyped with Bovine HD Bead Chips followed by estimation of FROH and ROH island using PLINK 1.9 program. Overall, the manuscript data is inline with the conclusions/interpretation. There are some issues with the structure/organization of manuscript that would help for the better readability.
1. Abstract- The gap/lacunae and the rationale are not clear in the abstract.
R/ Was the paragraph rewritten for better understanding.
- Flow of information in the abstract in not well organized. It requires revision.
R/ The abstract was revised.
- Line 14. “ROH analyzes can help reveal genetic relationships……”. ROH analyzes???
R/ The correction was made in the text.
- Line 21-22. “We found genomic regions on chromosomes BTA3, BTA5, BTA6, BTA8 and BTA16”. What kind of genomic regions?? This sentence needs to be revised.
R/ The correction was made in the text.
- Line 147. Please revise the title of this subheading of the result section. In current form it is not informative. For instance, "Genetic population structure of CAQ exhibits low population stratifications.
R/ Was inserted into the text.
- Figure 2 ligand is not explanatory. Please consider explaining the graph components.
R/ The figure was explained in the legend
- Line 170. Please consider revising this title heading. "Runs of homozygosity" is not informative.
R/ The line was rewritten for better informative
- Line 185. Similar to the above points, please consider revising the title of the heading.
R/ The line was rewritten for better informative
- Both Results and Discussion sections have used the same heading titles. For the sale of readability and understanding, I would recommend using some informative message containing titles for these heading.
R/ Were revised the heading titles and rewritten for better understanding.
Reviewer 2 Report
Introduction.
Please include a paragraph with details of the breed (anatomic details, production outcomes, etc.) and also please include some photographs of male and female individuals of the breed.
In a final paragraph, please summarize the hypothesis and please present clearly the objectives of the study.
M & M
2.2. Briefly explain the validity of using this equation in the present work.
2.3. Briefly explain the validity of using this coefficient in the present work.
Results.
Figure 2. Please replace with a sharp diagram.
Table 1 must be presented also as a figure to allow readers a quick appraisal.
Discussion
Please include an initial sub-section as 4.1., titled preface or something similar.
Please note that several relevant references are missing and must be included in the corrected version.
Overall.
The manuscript can advance to the correction stage, but really requires a lot of improvement to become acceptable. Please follow carefully the above points and proceed to correct them before resubmitting.
Author Response
Thank for the suggestions on the manuscript. The paragraphs that were revised are highlighted in yellow.
The version revised manuscript has undergone English language editing by MDPI.
Introduction.
Please include a paragraph with details of the breed (anatomic details, production outcomes, etc.) and also please include some photographs of male and female individuals of the breed.
R/ Was inserted into the text.
In a final paragraph, please summarize the hypothesis and please present clearly the objectives of the study.
R/ Was the paragraph rewritten for better understanding.
M & M
2.2. Briefly explain the validity of using this equation in the present work.
R/ Was used identity by descent (IBD) estimates for all pairs of individuals; they detect sample contaminations, swaps, duplications, pedigree errors, and unknown familial relationships. The estimates of pairwise IBD to find pairs of individuals who look too similar to each other, i.e., more than we would expect by chance in a random sample.
2.3. Briefly explain the validity of using this coefficient in the present work.
R/ Was to the lack of genealogical information on the population, the inbreeding coefficient FROH was used, which only requires information on the genotype of the animals. The inbreeding estimated from runs of homozygosity (FROH) more accurately predicts the current autozygosity percentage of the genome and detects autozygosity due to a common ancestor even 50 generations earlier (Howrigan et al . 2011; Keller et al . 2011).
Results.
Figure 2. Please replace with a sharp diagram.
R/ Was replaced with a sharp diagram.
Table 1 must be presented also as a figure to allow readers a quick appraisal.
R/ Was include with figure.
Discussion
Please include an initial sub-section as 4.1., titled preface or something similar.
R/ Was inserted into the text the heading titles.
Please note that several relevant references are missing and must be included in the corrected version.
R/Was revised the references of manuscript
Round 2
Reviewer 1 Report
Authors have addressed my concerns in the revised version. Manuscript is improved. I recommend its publication.
Author Response
Thank you for the suggestions to improve the manuscript.
Reviewer 2 Report
The manuscript reads better now, after making the changes suggested.
There is however room for some further improvement. The justifications for using the equation in sub-section 2.2. and the coefficient in sub-section 2.3. should be inserted in the respective passages in the tests, along with the supporting references, in order to provide details to future readers about the methodology followed.
Author Response
There is however room for some further improvement. The justifications for using the equation in sub-section 2.2. and the coefficient in sub-section 2.3. should be inserted in the respective passages in the tests, along with the supporting references, in order to provide details to future readers about the methodology followed.
R/ Thank for the suggestions on the manuscript. Was inserted into the text the justifications for using the equation in sub-section 2.2. and the coefficient in sub-section 2.3. The paragraphs are highlighted in yellow.